# Effects of 60-Day *Saccharomyces boulardii* and Superoxide Dismutase Supplementation on Body Composition, Hunger Sensation, Pro/Antioxidant Ratio, Inflammation and Hormonal Lipo-Metabolic Biomarkers in Obese Adults: A Double-Blind, Placebo-Controlled Trial

**DOI:** 10.3390/nu13082512

**Published:** 2021-07-23

**Authors:** Mariangela Rondanelli, Niccolò Miraglia, Pietro Putignano, Ignazio Castagliuolo, Paola Brun, Stefano Dall’Acqua, Gabriella Peroni, Milena Anna Faliva, Maurizio Naso, Mara Nichetti, Vittoria Infantino, Simone Perna

**Affiliations:** 1Department of Public Health, IRCCS Mondino Foundation, 27100 Pavia, Italy; mariangela.rondanelli@unipv.it; 2Department of Public Health, Experimental and Forensic Medicine, University of Pavia, 27100 Pavia, Italy; viriainfantino@hotmail.it; 3Clinical & Pre-Clinical Development, Gnosis SpA, 20121 Milan, Italy; n.miraglia@gnosis.lesaffre.com; 4Business Unit of the Lesaffre Group, Lesaffre, 59703 Marcq-en-Baroeul, France; 5SP Diabetic Outpatient Clinic, ASST Monza, 20900 Monza, Italy; pputignano@virgilio.it; 6Department of Molecular Medicine, University of Padova, 35121 Padova, Italy; ignazio.castagliuolo@unipd.it (I.C.); paola.brun.1@unipd.it (P.B.); 7Department of Pharmaceutical and Pharmacological Sciences, University of Padova, 35131 Padova, Italy; stefano.dallacqua@unipd.it; 8Endocrinology and Nutrition Unit, Azienda di Servizi alla Persona “Istituto Santa Margherita”, University of Pavia, 27100 Pavia, Italy; milena.faliva@gmail.com (M.A.F.); mau.na.mn@gmail.com (M.N.); dietista.mara.nichetti@gmail.com (M.N.); 9Department of Biology, Sakhir Campus, College of Science, University of Bahrain, Sakhir 32038, Bahrain; simoneperna@hotmail.it

**Keywords:** *Saccharomyces boulardii*, superoxide dismutase, obesity

## Abstract

In animals it has been demonstrated that *Saccharomyces boulardii* and Superoxide Dismutase (SOD) decrease low-grade inflammation and that *S. boulardii* can also decrease adiposity. The purpose of this study was to evaluate the effect of a 60-day *S. boulardii* and SOD supplementation on circulating markers of inflammation, body composition, hunger sensation, pro/antioxidant ratio, hormonal, lipid profile, glucose, insulin and HOMA-IR, in obese adults (BMI 30–35 kg/m^2^). Twenty-five obese adults were randomly assigned to intervention (8/4 women/men, 57 ± 8 years) or Placebo (9/4 women/men, 50 ± 9 years). Intervention group showed a statistically significant (*p* < 0.05) decrease of body weight, BMI, fat mass, insulin, HOMA Index and uric acid. Patients in intervention and control groups showed a significant decrease (*p* < 0.05) of GLP-1. Intervention group showed an increase (*p* < 0.05) of Vitamin D as well. In conclusion, the 60-day *S. boulardii*-SOD supplementation in obese subjects determined a significant weight loss with consequent decrease on fat mass, with preservation of fat free mass. The decrease of HOMA index and uric acid, produced additional benefits in obesity management. The observed increase in vitamin D levels in treated group requires further investigation.

## 1. Introduction

Obesity is a clinical condition of excess body adiposity, and is associated with several risk factors, such as cardiovascular risk, diabetes and cancer [1].

The prevalence of obesity and excess weight in adults, but also in children and adolescents, has increased extensively in recent years, leading to high healthcare costs [2].

Obesity is a broad combination of energy intake and lifestyle; with these being factors relevant, it has been suggested that a sub-optimal intestinal microbiota can have an important influence, as well as inflammation [3,4].

Specifically, in pathological conditions such as high adiposity and type 2 diabetes, the gut microbiota can have a significant interaction with the host, leading to the development of associated metabolic disorders [5].

Experimental studies have shown that germ-free mice have 40% less fat mass compared to control mice [6]. However, a significant increase in fat mass is observed in germ-free mice following colonization with the microbiota of obese mice [7]. Overall, a growing body of experimental evidence in animal models and humans suggests that the gut microbiota is involved in the regulation of energy homeostasis, metabolic inflammation, an d lipid and glucose metabolism [8]. Therefore, it is not surprising that supplementation of rats fed on a carbohydrate-rich diet with the lactic acid producing bacteria *L. gasseri*, *L. casei* and *L. acidophilus* limited the weight increase [9].

Although it has also been established recently that *S. boulardii* has a positive effect on intestinal health in adults and children [10], its potential role in obesity, hepatic disorders and metabolic inflammation is still largely unknown [8]. In diabetic and leptin-resistant obese mice, *S. boulardii* supplementation for 4 weeks induced a significant decrease in body weight, as well as in fat mass measured by nuclear magnetic resonance, fat index and weight of adipose and epididymal visceral tissue [11].

Moreover, in mouse models of diet-induced obesity, *S. boulardii* supplementation decreases liver weight, lipids content, and markers of macrophage infiltration.

In addition, *S. boulardii* supplementation in mice alters the balance of major intestinal phyla inducing a significant increase in Bacteroides and a marked decrease in Firmicutes, Proteobacteria and Tenericutes [8].

In obesity, inflammation is sustained by the increased permeability of the intestinal mucosa, which promotes the passage of bacterial endotoxins in the systemic circulation [12]. These endotoxins can cause damage to the host by triggering inflammatory cytokines release. Endotoxins and inflammatory cytokines can cause insulin dysfunction that determines glucose intolerance, and therefore diabetes [13]. Among the soluble mediators released by activated macrophages are free oxygen radicals, which contribute to tissue damage by compromising the barrier function of the intestinal mucosa and further favoring the passage of endotoxins from the intestinal lumen [14]. Among the various cellular systems that can eliminate free radicals, such as the O2-an anion, there is Superoxide Dismutase (SOD). SOD is one of the most relevant enzymes in the organism involved in reducing reactive oxygen species (ROS). ROS increase is associated with obesity and is induced by the inflammatory cytokines produced by adipose tissue [15]. SOD activity is significantly diminished as a result of an increase of adipose tissue [15]. Consequently, SOD administration has been used for the treatment of inflammatory diseases [16]. Moreover, it was demonstrated in animal models that after 1 month of supplementation with SOD, weight, adipose tissue and insulin resistance were reduced and hepatic oxidative stress was corrected [17].

Given this background, the association of *S. boulardii* with SOD in obesity management seems suitable. The purpose of this study was to assess the efficacy and safety of a 60-day *S. boulardii* and SOD supplementation in improving obese status. Inflammatory markers (CRP, TNF-α, IL-1β, IL-6), body composition (assessed by Dual Energy X-Ray Absorptiometry-DXA-and vector impedance analysis), empty sensation (assessed by visual-analogic scale and peptide YY), pro/antioxidant ratio (assessed by trimethylamine N-oxide, TMAO), hormonal (assessed by ghrelin, leptin, adiponectin and glucagon like peptide 1, GLP-1) and lipo-metabolic profile, such as cholesterol profile, triglycerides (TG), and glycemic profile, were determined in adults affected by obesity.

## 2. Materials and Methods

### 2.1. Design

This study was a mono-center, prospective, randomized, double blind, placebo-controlled clinical trial performed at Istituto Santa Margherita (University of Pavia). The experimental protocol was approved by the Ethics Committee of the University of Pavia (ethical code Number: 1209/151217) and was registered at ClinicalTrials.gov (NCT04919850).

### 2.2. Subjects

After screening for eligibility, twenty-five obese adults were enrolled and randomly assigned to the intervention group or the placebo (Figure 1).

Of the subjects who completed the study, 12 (8 females and 4 males) were assigned to the intervention group and 13 (9 females and 4 males) were assigned to the placebo group. The duration of treatment was 60 days. Screen visits were scheduled at time 0 (T0) and at the end of treatment (T1). Participants gave their written consent to this study. Obese men and women (BMI between 30 and 35 kg/m^2^), aged 30 to 65 years, admitted as outpatients at ASP Santa Margherita Hospital, Pavia, Italy were enrolled. Patients with clinical evidence of heart, kidney or liver disease were excluded.

Moreover, patients were excluded if they met the Diagnostic and Statistical Manual-IV (DSM-V) criteria for a current diagnosis of major depressive disorder as determined by the Structured Clinical Interview for DSM-V Axis 1 Disorders (SCID-1) [18].

Current medical drugs for weight loss, for control of cholesterol and triglycerides (TG), anti-inflammatory treatments, pregnancy or lactation were further exclusion criteria, as well as recent menopause manifestation. Other exclusion criteria were Type 1 diabetes mellitus, intestinal inflammatory bowel disease, celiac disease, chronic pancreatitis, antibiotic use within the last 3 months, and probiotic/prebiotic treatment in the last 4 weeks.

All participants agreed to avoid participating in any other weight loss diet treatment during this study. Alcohol intake, smoking habits and physical activity information were collected. Data were gathered from the end of January 2018 to the end of June 2020.

### 2.3. Dietary Treatment and Supplementation

In both groups, the subjects were trained to restrict their daily energy intake by a moderate amount, that is, 3344 kJ/day lower than the daily requirement based on World Health Organization criteria. The diet regime was based on a balance of macronutrients consisting of 25% of energy from fats, 60% of energy from carbohydrates and 15% of energy from proteins [19].

Subjects in the experimental group received gastro-resistant capsules containing a complex of 250 mg of the *Saccharomyces cerevisiae* variant *boulardii*, strain DBVPG 6763 (5.0 × 10^9^ CFU) and 500 IU SOD to be assumed twice/day for 60 days at mealtimes. The capsules were gastro-resistant to preserve SOD activity from gastric juice. *S. boulardii* was produced by Gnosis by Lesaffre, a Business Unit of the Lesaffre Group (Marcq-en-Baroeul, France), SOD was purchased by Phenbiox (Bologna, Italy). SuperOx-D, the SOD specialty by Phenbiox, is a Cu/Zn Superoxide dismutase of vegetal origin, extracted from carrots. Capsules also contained mannitol, microcrystalline cellulose 112 and silica as excipients. Capsules were coated with Eudagrad control^®^, a methacrylate approved for food, sodium alginate and talc micronized; all these coating ingredients conferred gastro-resistance to the capsules. The control group received a placebo consisting of capsules containing the same excipients except the active compounds, and the same coating.

### 2.4. Body Composition and Visceral Adipose Tissue (VAT) Assessment

Body weight and body composition (fat mass, visceral fat mass and free fat mass) parameters were obtained through the DXA (Lunar Prodigy DXA, GE Healthcare Medical Systems) [20].

### 2.5. Biochemical Analyses

Fasting serum glucose was assessed with a Kodak EKtachem 700 Analyzer (Eastman Kodak, Rochester, NY, USA). The quantity of high-density lipoprotein cholesterol (HDL) was determined using an electrophoretic and densitometric method (REP Lipoprotein-Kin; Helena Diagnostics, Gateshead, UK). Regarding the triglycerides assay, the Peridochrom Triglyceride GPO-PAP Kit was used (Boehringer, Mannheim, Germany). For the total cholesterol assay the CHOD-PAP Kit (Boehringer) was used. Serum insulin was evaluated by a double antibody RIA (Kabi Pharmacia Diagnostics AB, Uppsala, Sweden) and expressed as microU/mL.

Insulin resistance was assessed using the homeostasis model assessment (HOMA) [21]. Total proteins, total bilirubin, iron, uric acid, creatinine, transaminase alanine aminotransferase, aspartate aminotransferase and gamma glutamyl transferase were measured using enzymatic-colorimetric methods [22]. The CRP was determined by Nephelometric High-Sensitivity CRP (Dade Behring, Marburg, Germany). Hemochrome was measured using a Coulter automated cell counter MAX-M (Beckman Coulter, Inc., Fullerton, CL, USA).

Plasma acylated ghrelin levels were measured using an enzyme immunometric assay based on a double-antibody sandwich technique (BioVendor, Brno, Czech Republic). Serum adiponectin levels were measured using an enzyme-linked immunosorbent assay (ELISA) (R&D Systems, Inc., Minneapolis, MN, USA). Serum leptin levels were assessed using an ELISA (R&D Systems, Inc., Minneapolis, MN, USA). Human peptide YY levels were assessed in plasma using quantitative sandwich ELISA (abcam, VWR, Milan, Italy); the intra-assay coefficient of variation was 11.5%. Human Glucagon-like peptide 1 (GLP1) was quantified in plasma samples by specific ELISA kit (Invitrogen, ThermoFischer Sci, Monza, Italy) with inter- and intra-assay coefficient of variation of 3.6%.

### 2.6. Circulating Cytokines Evaluation

Plasma IL-6, TNF-α and IL1-β levels were measured with high-sensitivity enzyme-linked immunosorbent assays (ELISA) purchased from eBioscience (Prodotti Gianni, Milan, Italy). The overall coefficient of variation ranged from 3.8% to 4.6%. Horseradish peroxidase activity was detected using TMB chromogenic substrate. Optical density at 450 nm was measured using Tecan’s Sunrise absorbance microplate reader (Tecan, Männedorf, Switzerland).

### 2.7. Determination of TMA and TMAO

TMA and TMAO were determined in plasma samples by LC-MS/MS using a previously described protocol [23] with minor modifications. An LC system was equipped with a binary pump and an autosampler (Agilent Technologies, Palo Alto, CA, USA) model 1260. An Agilent Poroshell HILIC, 3.00 × 100, 2.7 µm was used as stationary phase, and water solutions of ammonium acetate (pH 4.00) 2.5 mM (A) and Acetonitrile (B) were used as mobile phases for a gradient elution. Gradient started with 20% B and increased to 50% B in 5 min at a flow rate of 0.4 mL/min. A triple quadrupole model 320 (Varian Inc., Palo Alto, CA, USA) equipped with Electrospray ion source working in positive ion mode was used as a detector. Plasma (500 mL) was deproteinized by adding a mixture containing an internal standard solution (Benzanilide) and methanol (19:1:60 vol:vol:vol). Solutions were centrifuged at 13,000 rpm and supernatant was collected. 20 μL of each supernatant was used for LC-MS/MS determination, transitions used for the quantification of the compounds were 60.2→44.7 (capillary 20; collision energy 12.0 V) for TMA, and 76.5→58.5 (capillary 30; collision energy 8.0 V) for TMAO, for the IS transition were 198.2→105.0 (capillary 50; collision energy 11.0 V). Limits of Quantification were 0.90 µM and 0.40 µM for TMA and TMAO, respectively.

### 2.8. Food Frequency Questionnaire (FFQ)

A validated Short-FFQ was used with the objective of analyzing food preferences of subjects at baseline and post-intervention. In particular, participants were asked by a dietician about their consumption of 18 common food items, choosing one between “yes” or “no, never”. Participants were also asked to estimate their usual rate of consumption, choosing from seven categories of frequency, ranging from “never” or “less than once a week” to “seven times per week”. The only exception was constituted by the item “coffee consumption”, which is reported as “cups/day”. The Short-FFQ used in this study was validated together with the Medium-FFQ (36 items) in Italian and then translated into English [24].

### 2.9. Eating Motivation Visual Analogue Scale (VAS)

Every day, in the evening, after dinner and before going to sleep, hunger, fullness, desire to eat, satiety and prospective food consumption were recorded by using a 100 mm Visual Analogue Scale (VAS). Each scale represents one of these five aspects and is recorded from a minimum of 0 to a maximum of 10 points. The scale of hunger (question: “how hungry are you?”) ranged from the answer “not at all” to “as hungry as I have ever felt”; the scale of fullness (question: “how full are you?”) ranged from the answer “not at all” to “as full as I ever felt”; the scale of satiety (question: “how satiated are you?”) ranged from “not at all” to “extremely”; the scale of desire (question: “how strong is your desire to eat?”) ranged from “very weak” to “very strong”; the scale of prospective consumption (question: “how much do you think you could (or would want to) eat right now) ranged from “nothing at all” to “a very large amount”. The characteristics of these scales are the same as in the questionnaire by Blundell et al., from which questions were translated into Italian [25]. The reliability and validity of VAS as a tool for assessing motivation to eat in humans were reviewed by Stubbs et al. in 2000 [26].

### 2.10. Beck Depression Inventory Questionnaire (BQ)

BQ was developed in 1961 as an instrument designed to measure the behavioral manifestation of depression. This depression inventory can be self-scored. The questionnaire is divided into 21 items, which have a score from 0 to 3. A total score >17 indicates borderline depression; >21 moderate depression; >31 severe depression; >40 extreme depression [27].

### 2.11. Safety

Safety was assessed by the measurement of blood pressure and by routine blood biochemistry parameters at T0 and T1 and by recording all the adverse events (AEs), which are based on spontaneous reporting by subjects as well as opened enquiries by members of the research staff.

### 2.12. Statistical Analysis

The sample size, determined as 30 patients (15 patients in each group) was set, with a difference of 4.1 kg, standard deviation (SD) of 1.2, two-sided two-sample *t*-test at 0.05 level of significance, dropout rate of 10%, and 90% power according to the study of Sharafedtinov et al. in 2013 [28].

The statistical analysis and reporting of this study were conducted in accordance with the CONSORT guidelines, with the primary analysis based on the full analysis set. For the baseline variables, the summary statistics employed frequencies and proportions for categorical data, and mean and SD for continuous variables.

Baseline variables were compared using the chi-square test for categorical outcomes and unpaired *t*-tests for continuous variables, as appropriate.

In the primary analysis, the baseline-adjusted means and 95% confidence interval (CI) estimated by ANCOVA with the change in primary outcomes were compared intra-group and between the placebo and intervention groups (intervention–placebo). The comparisons were adjusted for age, gender and baseline values of the variables.

ANCOVA were used for the secondary outcomes at each time point. All *p*-values were two-sided *p*-values.

## 3. Results

Twenty-five obese adults were recruited and randomly assigned to intervention or placebo groups (Figure 1). All subjects completed the study, 12 (8 women, 4 men) in the intervention group and 13 (9 women, 4 men) in the placebo group.

There were no significant differences at baseline between the subjects of the two groups for any of the variables analyzed (Table 1).

Table 2 shows the effects of treatment and placebo after 60 days of supplementation. In the table, the differences in the observed parameters are reported between the value at the end of the supplementation (T1) and at baseline (T0). Negative values represent a decrease and positive values an increase of the considered parameters during treatment. The intervention group showed a statistically significant (*p* < 0.05) decrease of body weight (Δ = −2.73 kg, CI: −4.72; −0.74), while no significant differences were seen in the placebo group between the end of treatment and baseline (T0). Additionally, BMI index results were significantly reduced by supplementation (Δ = −0.97 kg/m^2^, CI: −1.70; −0.25), while remaining unaffected in the placebo. Concerning body composition, a reduction in fat mass was demonstrated after 60 days of supplementation (DXA) (Δ = −3.13 kg, CI: −5.22; −1.05), but not in the group assuming the placebo (Table 2). No significant changes were observed in fat-free mass in either group, while 60 days of supplementation resulted in a significant reduction in android fat only in the intervention group. Taking into consideration hematological metabolic parameters, insulin (Δ = −2.27 mcU/mL, CI: −3.83; −0.71), Homa Index (Δ = −0.48 pts, CI: −0.87; −0.09) and uric acid (Δ = −0.50 mg/dL, CI: −0.98; −0.02) showed a significant decrease after 60 days of *S. boulardii*/SOD supplementation.

Intervention and placebo groups were contemporarily associated with a significant decrease (*p* < 0.05) of GLP-1 (Δ = −2.39 pmol/L, CI: −3.27; −1.51 and Δ = −2.05 pmol/L, CI: −2.89; −1.21, respectively).

Placebo alone was associated with a statistically significant decrease (*p* < 0.05) of circulating levels of ALT (Δ = −3.39 U/L, CI: −5.94; −0.83) and total ghrelin (Δ = −1.28 pg/mL, CI: −2.39; −0.18), while the intervention group showed a statistically significant increase (*p* < 0.05) in Vitamin D circulating levels (Δ = 5.22 ng/mL, CI: 1.56; 8.88).

Despite many parameters indicating a significant change in the intervention group after supplementation with the *S. boulardii*/SOD composition, statistical analysis showed no statistically significant effects when comparing the intervention to the placebo group (Treatment A–Treatment B) (Table 2). This may be due to individual variability related to the small number of subjects involved.

Close to statistically significant *p*-values were found for TMAO circulating levels, which were found to be decreased (*p* = ns) in the treatment group after 60 days and increased (*p* = ns) in the placebo group (Δ between groups TMAO = −2.78 µM, *p* = 0.083).

On the other hand, total cholesterol and LDL cholesterol circulating levels were raised in the intervention group, even if this was not statistically significant (Δ between groups Total Cholesterol: = +11.69 mg/dL, *p* = 0.065; Δ between groups LDL cholesterol = +12.15 mg/dL, *p* = 0.084).

Figure 2 shows the 18 answers taken from the Short-Form FFQ. In the intervention group, statistically significant differences were observed for the frequency of consumption of skimmed dairy products (increased), yoghurt (both portion and frequency decreased), and white meat (increased) over the week. In the Placebo group, a significant decrease of weekly bread consumption was observed.

Regarding Eating Motivation VAS, this test was performed to examine the significant differences between groups. No statistically significant differences were observed in any of the aspects of Eating Motivation (*p* = ns); data are shown in Figure 3.

## 4. Discussion

The effect of a supplementation of *S. boulardii* and SOD on physical and metabolic parameters in obese subjects was investigated in this study.

### 4.1. Effects of Supplementation on Body Composition

The present study demonstrates that 60-day intake of a dietary supplement consisting of an association of *S. boulardii* and SOD in a group of subjects affected by grade 1 obesity induced a significant reduction in body weight and BMI compared to baseline, while no changes occurred in the placebo-treated group, even if no statistically significant inter-group differences were observed. The modification of the fat mass induced by *S. boulardii*/SOD supplementation is of particular interest: a significant decrease in fat mass was observed after supplementation, while, at the same time, free fat mass did not change significantly, as shown by the results of the DXA evaluation. In the placebo group, no statistically significant variation was demonstrated for these parameters.

### 4.2. Effects of Supplementation on Metabolism

From a clinical point of view, it is interesting to note the reduction of insulin levels and insulin resistance, which are reflected in the reduction of the HOMA index.

This clinical trial confirmed in humans what had already been shown in animal model: *S. boulardii* metabolic activities could be beneficial to patients with obesity, even if the mechanism of action of this yeast in the gut remains to be clearly defined.

To date, no probiotic yeast has been investigated in the context of obesity in humans. A recent study showed that type 2 diabetic and obese mice (db/db) treated with *S. boulardii* exhibited reduced body weight, fat mass, hepatic steatosis, and inflammatory tone [8], concluding that *S. boulardii* induced a modification of the gut microbiota composition that was related to a change in host metabolism responses. Growing evidence supports that gut microbiota–host interactions control energy homeostasis, glucose metabolism, and lipid metabolism [29,30,31,32]. Therefore, strategies able to modify the gut microbiota in the context of obesity could be a way to treat these pathologies, and the use of probiotic bacteria has also been suggested [31,33,34,35,36,37,38,39].

### 4.3. Effects of Supplementation on Vitamin D Levels

A promising and innovative result of this study concerns the increase in vitamin D levels in the treated group [40].

Recently, probiotics have been shown to be effective supplements for improving vitamin D deficiency. Studies reported that the use of probiotics is related to enhanced vitamin D absorption, and to increased vitamin D receptor (VDR) expression [41,42]. Interestingly, a clinical study reported a significant increase in serum 25-hydroxyvitamin D_3_ levels with oral supplementation of probiotic *Lactobacillus reuteri* strain [43].

Overall, these studies support the view that gut microbiota is a potential nutritional and pharmacological target in the management of obesity and obesity-related disorders, such as vitamin D deficiency [44].

### 4.4. Effects of Supplementation on Uric Acid Levels

A further interesting result of this study is related to the decrease in uric acid levels detected only in the treated group.

It has been demonstrated, using mouse models, that adipose tissue can produce and secrete uric acid through xanthine oxidoreductase (XOR) and that this production is enhanced in obesity [45].

An elevated serum uric acid level (hyperuricemia) is closely associated with visceral fat accumulation [46,47,48] and with various metabolic disorders, such as glucose intolerance, elevated blood pressure, dyslipidemia, and atherosclerotic cardiovascular diseases, conceptualized as metabolic syndrome [49,50,51,52,53].

Given this background, we propose that the decrease of uric acid levels, noticed in treated patients but not in the placebo group, be due to the reduction of adipose tissue as previously demonstrated [54]. Therefore, in line with previous studies, we suggest that the decrease in uric acid levels detected in treated patients, although levels are in the normal range, could be due to the decrease in adipose tissue [54]. Moreover, the presence of SOD in the dietary supplement could be a valuable support in decreasing uric acid levels. Further investigation is required to demonstrate these hypotheses.

### 4.5. Role of Superoxide Dismutase

SOD supplementation could have a possible pivotal role in decreasing body weight, adipose tissue weight and insulin resistance by activating lipolysis in adipocytes and thus reducing their size, as demonstrated in animal models [17].

Regarding the hypoxia of obese adipose tissue, many studies have reported that, in obese patients, hypoxia induces dysfunction of adipose tissue, such as dysregulation of adipocytokines and chronic inflammation [55], which could be rescued by superoxide dismutase [56].

### 4.6. Effects of Supplementation on Other Parameters

The reduction of GLP-1 in both groups is difficult to interpret, although we should probably consider it to be like a response to the reduction of nutrient intake [57]. The reduction in ghrelin levels, found only in the placebo group, could also be explained in the same way [58].

All the other parameters studied showed no significant T1–T0 differences within the groups nor a significant difference between the groups. This is presumably due to the low number of patients and to the fact that the patients studied may belong to the group of metabolically healthy obese (MHO) subjects, as defined by Karelis [59].

The present study has some limitations. The first limitation is that the results demonstrated no significant differences between treated groups and control groups after intervention. Moreover, given the small sample size and the brief follow-up duration in this study, generalization of our findings to the larger population of obese people should be approached with caution. Finally, provided the small sample size, it was not possible to carry out a statistical analysis considering menopausal women versus younger men or vice versa and considering gender separation, which may influence the observed differences in cholesterol levels.

Despite these limitations, our study design has strength in allowing for an exact analysis of body composition by DXA, providing robust evidence for the action of dietary supplementation controlling adipose tissue distribution in obesity. Considering most weight loss studies involve treatments ranging from 12 weeks up to 2–3 years, further studies are required to evaluate the long-term efficacy of *S. boulardii* and SOD in a larger and more diverse patient population, as well as in patients with obesity-related diseases such as type 2 diabetes.

## 5. Conclusions

In conclusion, this study demonstrated that 60-day *S. boulardii* and SOD supplementation in obese subjects determined a significant reduction in BMI, with weight loss due to the decrease in fat mass, with preservation of fat free mass, HOMA index and uric acid, suggesting an interesting and promising role of the supplementation with *S. boulardii* in association with SOD in treating obesity.

## Figures and Tables

**Figure 1 nutrients-13-02512-f001:**
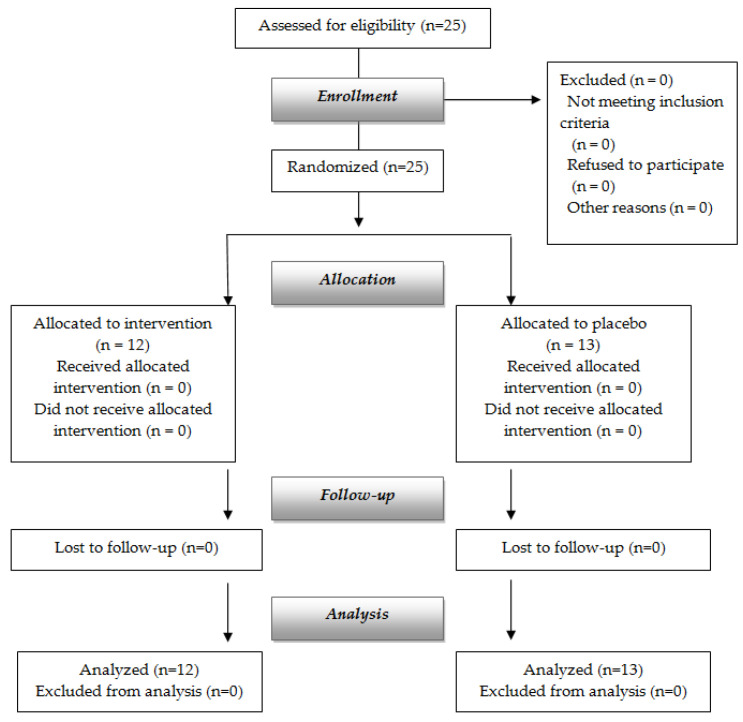
Flow chart of the study.

**Figure 2 nutrients-13-02512-f002:**
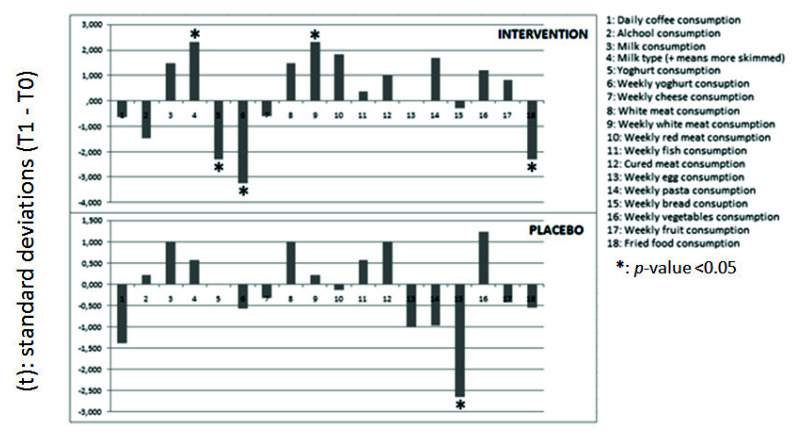
Food Frequency Questionnaire (FFQ) changes between T1 and T0.

**Figure 3 nutrients-13-02512-f003:**
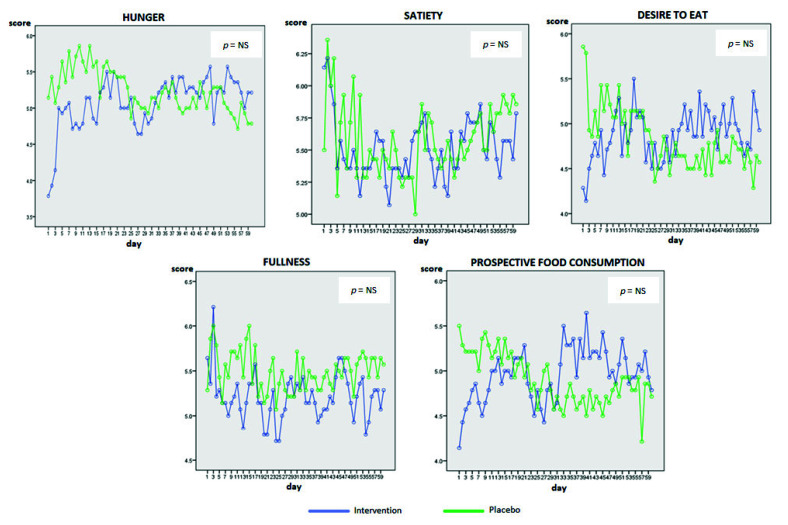
Graphical representation of Eating Motivation VAS changes in groups during 60 days.

**Table 1 nutrients-13-02512-t001:** Baseline descriptive characteristics of sample.

Variables	Intervention (Mean Values ± SD)	Placebo (Mean Values ± SD)	Total (Mean Values ± SD)	*p*-Value
**General characteristics**				
Gender (women/men)	8/4	9/4	17/8	Χ^2^ = 0.019 *p* = 0.891
Age	57.33 ± 8.86	50.31 ± 9.54	53.68 ± 9.71	0.644
**Body composition by DXA**				
Free Fat Mass (kg)	51.31 ± 7.60	51.42 ± 8.31	51.37 ± 7.81	0.569
Fat Mass (kg)	44.31 ± 10.05	43.65 ± 9.84	43.96 ± 9.72	0.922
Android fat (%)	51.71 ± 7.28	52.29 ± 7.33	52.01 ± 7.16	0.921
Gynoid fat (%)	45.42 ± 9.99	47.24 ± 8.93	46.36 ± 9.30	0.520
VAT (g)	1838.33 ± 699.33	1844.69 ± 931.57	1841.64 ± 811.22	0.561
Weight (kg)	96.63 ± 12.38	97.83 ± 11.76	97.26 ± 11.82	0.993
**Anthropometric measures**				
Height (m)	1.67 ± 0.06	1.67 ± 0.07	1.67 ± 0.07	0.360
BMI (kg/m^2^)	34.60 ± 3.03	35.04 ± 3.17	34.83 ± 3.05	0.889
**Others**				
Homa index (points)	2.20 ± 0.86	2.02 ± 1.25	2.21 ± 1.22	0.704
Beck Questionnaire (points)	5.55 ± 4.91	7.58 ± 5.50	6.61 ± 5.21	0.401
**Blood tests**				
Total cholesterol (mg/dL)	206.08 ± 20.57	215.60 ± 18.15	210.41 ± 19.66	0.263
LDL cholesterol (mg/dL)	132.33 ± 17.57	137.70 ± 21.90	134.87 ± 19.43	0.546
HDL cholesterol (mg/dL)	59.09 ± 10.32	61.10 ± 18.27	60.05 ± 14.30	0.764
Triglycerides (mg/dL)	79.00 ± 28.20	93.33 ± 27.71	86.17 ± 28.11	0.781
Glycemia (mg/dL)	89.75 ± 14.57	88.54 ± 10.19	89.12 ± 12.23	0.270
Insulin (mcU/mL)	9.90 ± 3.91	8.81 ± 4.93	9.82 ± 4.86	0.573
Apolipoprotein A (mg/dL)	163.08 ± 31.90	168.27 ± 37.72	165.57 ± 34.09	0.221
Apolipoprotein B (mg/dL)	115.50 ± 27.08	121.00 ± 21.87	118.37 ± 22.92	0.669
AST (U/L)	19.50 ± 5.92	20.69 ± 5.76	20.12 ± 5.75	0.736
ALT (U/L)	21.17 ± 12.22	22.00 ± 9.63	21.60 ± 10.72	0.210
δGT (U/L)	18.52 ± 10.30	17.12 ± 5.67	17.82 ± 8.16	0.149
Creatinine (mg/dL)	0.68 ± 0.18	0.78 ± 0.16	0.74 ± 0.17	0.806
Total Proteins (g/dL)	7.00 ± 0.48	7.00 ± 0.49	7.00 ± 0.48	0.782
Albumin (g/dL)	4.41 ± 0.36	4.33 ± 0.39	4.37 ± 0.37	0.657
CRP (mg/dL)	0.10 ± 0.09	0.25 ± 0.25	0.18 ± 0.20	0.078
Uric acid (mg/dL)	4.21 ± 1.29	4.35 ± 1.08	4.28 ± 1.16	0.265
Fe (µg/dL)	79.50 ± 28.61	82.38 ± 38.66	81.00 ± 33.53	0.307
TSH (µU/dL)	1.58 ± 1.19	1.73 ± 1.33	1.65 ± 1.23	0.860
Leptin (ng/cL)	3.94 ± 3.20	4.41 ± 2.91	4.19 ± 3.00	0.811
Ghrelin (pg/mL)	6.40 ± 1.35	5.77 ± 1.84	6.07 ± 1.62	0.566
Adiponectin (pg/mL)	149.50 ± 127.72	125.38 ± 69.49	135.54 ± 89.45	0.263
Peptide YY (pg/mL)	267.58 ± 46.77	283.58 ± 74.46	275.90 ± 61.98	0.073
GLP-1 (pmol/L)	10.94 ± 23.18	7.30 ± 11.41	9.05 ± 17.74	0.311
TNF-α (pg/mL)	66.50 ± 110.18	40.20 ± 29.54	39.11 ± 22.58	0.185

**Table 2 nutrients-13-02512-t002:** Treatment effect from baseline analyzed using linear mixed models and treatment effect between groups.

Variables	Intervention	Placebo	Difference between Groups (Confidence Interval)	*p*-Value between Groups
	**Δ T1-T0 (confidence interval)**	**Δ T1-T0 (confidence interval)**		
**DXA**				
Free Fat Mass (kg)	0.04 (−0.85; 0.93)	0.20 (−0.65; 1.05)	−0.15 (−1.44; 1.13)	0.805
Fat Mass (kg)	−3.13 (−5.22; −1.05) *	−1.88 (−3.78; 0.02)	−1.25 (−4.21; 1.70)	0.385
Android fat (%)	−2.13 (−3.63; −0.63) *	−0.62 (−2.06; 0.82)	−1.51 (−3.67; 0.65)	0.160
Gynoid fat (%)	−1.01 (−2.40; 0.38)	−0.64 (−1.97; 0.69)	−0.37 (−2.37; 1.63)	0.702
VAT (g)	−132.16 (−349.27; 84.94)	−135.00 (−342.80; 72.80)	2.84 (−311.54; 317.22)	0.985
DXA Weight (kg)	−2.73 (−4.72; −0.74) *	−1.00 (−2.90; 0.90)	−1.73 (−4.60; 1.14)	0.223
**Anthropometric measures**				
BMI (kg/m^2^)	−0.97 (−1.70; −0.25) *	−0.36 (−1.05; 0.34)	−0.62 (−1.67; 0.43)	0.235
**Others**				
Homa index (points)	−0.48 (−0.87; −0.09) *	−0.16 (−0.06; 0.23)	−0.31 (−0.91; 0.28)	0.280
Beck Questionnaire (points)	−1.36 (−3.47; 0.75)	−0.59 (−2.59; 1.42)	−0.77 (−3.86; 2.32)	0.606
**Blood tests**				
Total cholesterol (mg/dL)	7.31 (−0.83; 15.46)	−4.38 (−13.35; 4.60)	11.69 (−0.82; 24.20) *	0.065
LDL cholesterol (mg/dL)	6.08 (−3.15; 15.32)	−6.07 (−15.79; 3.65)	12.15 (−1.80; 26.10)	0.083
HDL cholesterol (mg/dL)	0.75 (−1.91; 3.40)	0.58 (−2.22; 3.37)	0.17 (−3.86; 4.20)	0.930
Triglycerides (mg/dL)	2.27 (−10.30; 14.84)	3.84 (−8.73; 16.41)	−1.57 (−20.06; 16.92)	0.857
Glycemia (mg/dL)	1.49 (−2.86; 5.84)	−3.22 (−7.39; 0.94)	4.72 (−1.54; 10.97)	0.132
Insulin (mcU/mL)	−2.27 (−3.83; −0.71) *	−0.55 (−2.11; 1.01)	−1.72 (−4.09; 0.65)	0.145
Apolipoprotein A (mg/dL)	1.52 (−5.84; 8.87)	−1.75 (−9.46; 5.96)	3.26 (−7.81; 14.34)	0.544
Apolipoprotein B (mg/dL)	5.27 (−4.06; 14.59)	7.03 (−1.81; 15.87)	−1.76 (−15.40; 11.87)	0.787
AST (U/L)	−1.46 (−3.49; 0.57)	−0.88 (−2.83; 1.06)	−0.58 (−3.51; 2.35)	0.684
ALT (U/L)	−2.11 (−4.55; 0.34)	−3.39 (−5.94; −0.83) *	−1.28 (−4.95; 2.39)	0.475
δGT (U/L)	−1.49 (−2.99; 0.02)	0.01 (−1.49; 1.52)	−1.50 (−3.70; 0.70)	0.170
Creatinin (mg/dL)	−0.00 (−0.04; 0.04)	0.02 (−0.02; 0.06)	−0.02 (−0.08; 0.04)	0.566
Total Proteins (g/dL)	0.11 (−0.14; 0.36)	−0.18 (−0.44; 0.08)	0.29 (−0.09; 0.67)	0.131
Albumin (g/dL)	−0.01 (−0.21; 0.19)	−0.17 (−0.39; 0.04)	0.17 (−0.13; 0.47)	0.249
CRP (mg/dL)	0.01 (−0.09; 0.08)	−0.01 (−0.08; 0.09)	0.02 (−0.11; 0.15)	0.799
Uric acid (mg/dL)	−0.50 (−0.98; −0.02) *	−0.10 (−0.56; 0.36)	−0.40 (−1.09; 0.29)	0.240
Fe (µg/dL)	7.13 (−12.93; 27.19)	−1.97 (−21.17; 17.24)	9.10 (−19.81; 38.00)	0.519
TSH (µU/dL)	−0.30 (−0.59; 0.00)	−0.12 (−0.43; 0.20)	−0.18 (−0.64; 0.28)	0.420
Leptin (ng/cL)	−0.45 (−1.60; 0.71)	−0.24 (−1.34; 0.87)	−0.21 (−1.88; 1.46)	0.796
Ghrelin (pg/mL)	−0.48 (−1.64; 0.67)	−1.28 (−2.39; −0.18) *	0.80 (−0.88; 2.48)	0.331
Adiponectin (pg/mL)	7.68 (−73.60; 88.97)	−7.88 (−74.50; 58.73)	15.57 (−95.33; 126.46)	0.765
Peptide YY (pg/mL)	−15.12 (−39.56; 9.32)	4.03 (−19.38; 27.43)	−19.14 (−54.37; 16.08)	0.270
GLP-1 (pmol/L)	−2.39 (−3.27; −1.51) *	−2.05 (−2.89; −1.21) *	−0.34 (−1.60; 0.92)	0.576
TNF-α (pg/mL)	−5.46 (−14.88; 3.95)	−8.79 (−17.81; 0.23)	3.33 (−10.23; 16.89)	0.614
IL-1β (pg/mL)	−0.56 (−2.16; 1.03)	0.54 (−0.92; 2.00)	−1.11 (−3.35; 1.14)	0.315
IL-6 (pg/mL)	−3.84 (−58.70; 51.01)	−5.24 (−62.75; 52.26)	1.40 (−81.35; 84.15)	0.972
TMA (µM)	0.67 (−6.76; 8.10)	4.86 (−1.72; 11.45)	−4.19 (−14.48; 6.10)	0.395
TMAO (µM)	−1.42 (−3.54; 0.70)	1.36 (−0.87; 3.59)	−2.78 (−5.99; 0.42)	0.084
Vitamin D (ng/mL)	5.22 (1.56; 8.88) *	1.27 (−2.24; 4.77)	3.95 (−1.33; 9.24)	0.134

* statistically significant at *p* < 0.05.

## Data Availability

The data presented in this study are available in this article.

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
