# Peer review of "Effects of 60-Day Saccharomyces boulardii and Superoxide Dismutase Supplementation on Body Composition, Hunger Sensation, Pro/Antioxidant Ratio, Inflammation and Hormonal Lipo-Metabolic Biomarkers in Obese Adults: A Double-Blind, Placebo-Controlled Trial"

_nutrients, 2021, doi:10.3390/nu13082512_

Round 1
Reviewer 1 Report
I found all comments are well improved now, so no further comments are there.
Author Response
thanks.
Reviewer 2 Report
The authors have made a major effort to adapt the comments of the first version.
The discussion is clearer and the material and methods are clearer, although small details remain, such as the inclusion of the units of measure in
Agilent Poroshell HILIC, 3.00 x 100, 2,7 m was (Line 193) instead of 2,7 µm ??
Plasma (500 L) (Line 198) instead of 500mL ???
Author Response
The authors have made a major effort to adapt the comments of the first version.
The discussion is clearer and the material and methods are clearer, although small details remain, such as the inclusion of the units of measure in
Agilent Poroshell HILIC, 3.00 x 100, 2,7 m was (Line 193) instead of 2,7 µm ??
ANSWER: the correct unit of measure is µm.
Plasma (500 L) (Line 198) instead of 500mL ???
ANSWER: the correct unit of measure is mL.
Round 2
Reviewer 2 Report
Accept in present form
This manuscript is a resubmission of an earlier submission. The following is a list of the peer review reports and author responses from that submission.
Round 1
Reviewer 1 Report
The paper entitled “Effect of 60-days……” was reviewed, and I found interesting data on human intervention experiment. The paper is well written and easy to understand, however there several small comments on this article.
(1) There are significant differences of body weight, BMI … and so on between pre and post in treated groups, but not in control groups. However, there are no significant differences between treated groups and control groups after intervention. Please add some comments in limitation section.
(2) Related to above comment, there are almost significant differences in total cholesterol (P<0.065) and LDL cholesterol (P<0.083) between treated group and control group. And treated group increased these values, whereas control group decreased these values. You should add some comments on these results.
(3) In lines 388-391, you discuss about GLP-1 reduction by nutrition intake, and also ghrelin reduction in control group by nutrition intake. But this discussion is lack of detailed explanation. For example, ghrelin reduction is related to reduction of bread intake (Fig.2)?
GLP-1 reduction is explained by which nutrient in Fig.2?
Reviewer 2 Report
The aim of the study is to assess the efficacy and safety of a 60-days S. boulardii and SOD supplementation in improving obese status
Introduction provides sufficient background and includes relevant references.
Search strategy, eligibility, study selection criteria, and data extraction process seems appropriate. The inclusion and exclusion criteria are the main strengths of the study
The weak point of the study is the number of participants, is very small and there is no gender separation, which may influence the observed differences in cholesterol levels. Menopausal women versus younger men or vice versa.
In humans there are two types of SOD, the first is located in the cell cytoplasm and its prosthetic group has Cu and Zn atoms (SOD Cu/Zn), while the second type is located in the mitochondria and has Mn. The authors do not indicate which type of superoxide dismutase they use.
TMA and TMAO were determined in plasma samples by LC-MS/MS using a pre-202 viously described protocol (Grinberga et al., 2015) which is not included in the references
To assess empty sensation, the authors analyse VAS and PYY. The authors state that no statistically significant differences are observed in any of the aspects of eating motivation (Figure 3). Between days 32 and 50, in terms of desire to eat and prospective food consumption, small behaviour differences are observed in the groups. This fact may be due because in the baseline descriptive characteristics of the sample, there was a slight increase in PYY in the placebo group with respect to the intervention group.
The role of SOD in the study is not sufficiently discussed. It does not even relate to TMAO. It is related only to uric acid levels.
The authors should homogenize the treatment time. In some places, it says 60 days (title) in others 8 weeks (Line 324-discussion).
In line 205 and line 210 the units of measurement are missing.
To aid understanding of the text, authors should include a table of abbreviations. i. e. Line 92 “.. assessed by DXA… Description for first time in line 146
English language and style are fine spell check required